# Clustering of Environmental Parameters and the Risk of Acute Myocardial Infarction

**DOI:** 10.3390/ijerph19148476

**Published:** 2022-07-11

**Authors:** Geraldine P. Y. Koo, Huili Zheng, Pin Pin Pek, Fintan Hughes, Shir Lynn Lim, Jun Wei Yeo, Marcus E. H. Ong, Andrew F. W. Ho

**Affiliations:** 1Ministry of Health Holdings, Singapore 099253, Singapore; geraldine.koo@mohh.com.sg; 2National Registry of Diseases Office, Health Promotion Board, Singapore 168937, Singapore; zheng_huili@hpb.gov.sg; 3Health Services & Systems Research, Duke-NUS Medical School, Singapore 169857, Singapore; maeve.pek@duke-nus.edu.sg (P.P.P.); marcus.ong@duke-nus.edu.sg (M.E.H.O.); 4Department of Anesthesiology, Duke University Hospital, Duke University, Durham, NC 27710, USA; fintan.hughes@duke.edu; 5Department of Cardiology, National University Heart Centre Singapore, Singapore 119074, Singapore; mdclims@nus.edu.sg; 6Department of Medicine, National University Singapore, Singapore 119228, Singapore; 7Yong Loo Lin School of Medicine, National University of Singapore, Singapore 119228, Singapore; yeojunwei1999@gmail.com; 8Department of Emergency Medicine, Singapore General Hospital, Singapore 169608, Singapore; 9Pre-Hospital and Emergency Research Centre, Duke-NUS Medical School Singapore, Singapore 169857, Singapore

**Keywords:** myocardial infarction, air pollution, haze, environmental epidemiology, clustering

## Abstract

The association between days with similar environmental parameters and cardiovascular events is unknown. We investigate the association between clusters of environmental parameters and acute myocardial infarction (AMI) risk in Singapore. Using k-means clustering and conditional Poisson models, we grouped calendar days from 2010 to 2015 based on rainfall, temperature, wind speed and the Pollutant Standards Index (PSI) and compared the incidence rate ratios (IRR) of AMI across the clusters using a time-stratified case-crossover design. Three distinct clusters were formed with Cluster 1 having high wind speed, Cluster 2 high rainfall, and Cluster 3 high temperature and PSI. Compared to Cluster 1, Cluster 3 had a higher AMI incidence with IRR 1.04 (95% confidence interval 1.01–1.07), but no significant difference was found between Cluster 1 and Cluster 2. Subgroup analyses showed that increased AMI incidence was significant only among those with age ≥65, male, non-smokers, non-ST elevation AMI (NSTEMI), history of hyperlipidemia and no history of ischemic heart disease, diabetes or hypertension. In conclusion, we found that AMI incidence, especially NSTEMI, is likely to be higher on days with high temperature and PSI. These findings have public health implications for AMI prevention and emergency health services delivery during the seasonal Southeast Asian transboundary haze.

## 1. Introduction

Cardiovascular disease remains a leading cause of morbidity and mortality worldwide [1]. There is growing evidence that environmental factors, such as temperature and air pollution, may play a larger role in mortality than previously accounted for [2,3]. Recent studies have found that particulate matter (PM) exposure was related to all-cause mortality and especially so for cardiovascular mortality [4,5]. The Global Burden of Disease Study identified ambient particulate matter as the 8th leading cause of death in the world [6]. There is also substantial evidence that demonstrates the effects of weather on mortality [7]. In particular, temperature has a possible non-linear, U or J-shaped effect on cardiovascular mortality and morbidity [8,9,10]. Hence, environmental exposure may be targeted to reduce the global burden of cardiovascular disease [6,11].

In Southeast Asia, transboundary haze is an annual occurrence with extensive public health and economic implications [12]. It is largely the result of seasonal, uncontrolled agricultural practices in Indonesia that use fire to clear land for vegetation farming, both at the small ‘slash and burn’ and large industrial scales [13,14]. The smoke created from such practices affect Indonesia and the neighbouring countries of Singapore, Malaysia, Brunei, reaching even as far as Southern parts of Thailand and the Philippines. Air quality and pollutant concentrations during the transboundary haze, especially of PM2.5 and O_3_, often rise to hazardous levels in excess of WHO guidelines [15,16]. The transboundary haze usually coincides with the dry season from September to October, with the El Nino–Southern Oscillation causing dry weather conditions and the positive Indian Ocean Dipole affecting air quality [12,13,15,17]. These have widespread impact on cardiovascular health and mortality in the region [12].

As Singapore is a small, densely populated urban city-state, there was no wide disparity in exposure to the transboundary haze among its population. Robust monitoring of weather parameters and close proximity of the monitoring stations to the population reduces the variation in the weather data collected and increases the accuracy in measuring the population’s exposure to the weather parameters. This makes Singapore an ideal population-based natural laboratory to study short-intermediate health impacts from the seasonal Southeast Asian haze [12].

This study aimed to identify distinct clusters of days based on weather and air quality conditions and determine their association with AMI incidence.

## 2. Materials and Methods

### 2.1. Setting

Singapore is a highly urbanised and densely populated city-state in Southeast Asia with a population of 5.4 million, well spread out throughout its land space of about 730 km^2^, and an average life expectancy of 83.9 years [18]. Singapore is situated near the equator, at the tip of the Malaysian Peninsula. It experiences largely uniform temperatures with abundant rainfall and humidity all year round. However, it does have two monsoon seasons—the Northeast Monsoon from December to March and the Southwest Monsoon from June to September [19].

The crude incidence rate of AMI in Singapore was 368.2 per 100,000 population in 2019, with a crude AMI mortality rate of 28.7 per 100,000 population in 2019 [20].

### 2.2. Study Population

The Singapore Myocardial Infarction Registry (SMIR) is a nationwide registry managed by the National Registry of Disease Office, Singapore [20]. It collects demographic and clinical data of AMI cases diagnosed in all public and private hospitals and out-of-hospital AMI deaths certified by medical practitioners in Singapore. The National Registry of Disease Act enacted in 2012 legally mandates notification of AMI cases to the SMIR.

The registry receives case notification from various sources, including patient medical claim listings, hospital in-patient discharge summaries, cardiac biomarker listings and the national death registry. The International Classification of Diseases, Ninth Revision, Clinical Modification (ICD-9-CM) code 410 was used to identify AMI cases diagnosed before 2012, while the ICD-10 (Australian Modification) codes I21 and I22 were used to identify AMI cases diagnosed from 2012 onwards.

Detailed data of each AMI patient were extracted from their clinical medical records, including ambulance records, emergency department notes, clinical charts and discharge summaries, by the registry coordinators. The data collected were audited yearly to ensure accuracy and inter-rater reliability.

The outcome variable of this study is the daily incidence of AMI in Singapore from 2010 to 2015. Recurrent AMI after 28 days of a recorded AMI episode was considered a separate episode [21]. The differentiation between ST-segment-elevation myocardial infarction (STEMI) and non-STEMI (NSTEMI) was based on presenting symptoms, cardiac biomarkers and ECG assessment, aligned with the clinician’s diagnosis as documented in the physical case notes and electronic medical records. STEMI was defined as typical chest pain of 30 min and significant ST segment elevation (0.1 mV or 0.2 mV on 2 adjacent limb or precordial leads, respectively, or new left bundle-branch block) and confirmed subsequently by a rise in cardiac biomarkers. All ECGs were interpreted, and all diagnoses adjudicated centrally at the National Registry of Disease Office.

We excluded patients with AMI that occurred during their hospital stay as these patients were admitted for non-AMI conditions initially and their AMI might have different pathophysiological mechanisms from the AMI of the other patients that occurred out-of-hospital. Event date was taken to be the date of AMI onset, which is the earliest reported start of acute symptoms, serial changes in ECG, elevation in cardiac biomarkers or fatal collapse.

### 2.3. Environmental Data

The 24-h Pollutant Standards Index (PSI) is used to indicate the level of pollutants in the air and report air quality in Singapore. It is computed based on 6 pollutants and devised by the United States Environmental Protection Agency: particulate matter (PM10), fine particulate matter (PM2.5), sulphur dioxide (SO_2_), nitrogen dioxide (NO_2_), ozone (O_3_) and carbon monoxide (CO). For each pollutant, a sub-index is calculated from a segmented linear function that transforms ambient concentrations into a scale extending from 0 through 500. PSI is the maximum of the 6 sub-indices. The National Environment Agency (NEA) of Singapore classifies 24-h PSI into ranges of good (0–50), moderate (51–100), unhealthy (101–200), very unhealthy (201–300) and hazardous (>300). The pollutants are measured from more than 20 telemetric air quality monitoring stations across the island [22]. Publicly available PSI data were retrieved from the NEA website. However, data on the 6 pollutants were not available. Publicly available weather data, such as daily temperature, rainfall and wind speed, were retrieved from the Meteorological Service Singapore’s website. The weather parameters are measured from 5 meteorological stations scattered across Singapore. The exposure variables of this study are daily mean temperature, total rainfall, mean wind speed and mean PSI in Singapore from 2010 to 2015.

### 2.4. Statistical Analysis

Daily mean temperature, total rainfall, mean wind speed and mean PSI were determined a priori for classifying the 2191 calendar days within the study period of 2010 to 2015 into clusters. A two-step cluster analysis was performed, in which the log-likelihood distance measure was used to separate the days into groups and the Akaike’s and Bayesian information criteria were used to choose the optimal number of clusters. The two-step clustering approach was chosen over the k-means and hierarchical clustering approaches as it does not require the number of clusters to be pre-defined and both numeric and ordinal variables can be used to form the clusters. Clusters formed using the weather and air quality data as numeric variables were less balanced in terms of sample size than when they were used as ordinal variables. Hence, daily mean temperature (<27.5 degrees Celsius, ≥27.5 degrees Celsius to <28.5 degrees Celsius, ≥28.5 degrees Celsius), total rainfall (0 mm, >0 mm to <2 mm, ≥2 mm), mean wind speed (0 to <7 km/h, ≥7 km/h to <10 km/h, >10 km/h) and mean PSI (0 to ≤50, >50 to ≤100, >100) were categorised into ordinal variables for the two-step clustering which divided the days within the study period into three clusters. Days with any missing environmental data were excluded from the cluster analysis. The three clusters were compared to see if there was any significant difference in weather parameters and air quality, as well as patients’ demographics and AMI clinical characteristics, using the Kruskal–Wallis test.

A time-stratified case-crossover approach was used to examine the association between each cluster and incidence of AMI. A case is a day with at least one AMI incidence and the case period was the day of AMI onset. Each case serves as its own control and the control periods were derived from the same month and year for the same days of the week as the AMI onset date of the corresponding case. Conditional Poisson regression was used to compare the incidence rate ratio (IRR) of AMI for each cluster between the case period and control period, accounting for overdispersion and autocorrelation [23]. We assessed the relationship between AMI incidence and clusters in the entire cohort and in subgroups of individual-level characteristics determined a priori. Statistical analyses were performed using STATA SE 13.

## 3. Results

### 3.1. Environmental Clusters

Temperature was positively correlated (Pearson correlation = 0.32) with PSI, while there was no significant correlation between PSI and rainfall (Pearson correlation = −0.096) and wind speed (Pearson correlation = 0.088) (Figure 1). Table 1 shows the characteristics of the three clusters. Generally, the days in Cluster 1 had higher wind speed (median 10.4 km/h; interquartile range (IQR) 8.4–12.2) than the other two clusters, the days in Cluster 2 had higher rainfall (median 2.8 mm; IQR 0.4–13.6), while the days in Cluster 3 had higher temperature (median 28.6 degree C; IQR 27.8–29.0) and higher PSI (median 59.2; IQR 54.8–70.0).

### 3.2. Study Population

There were 39287 episodes of AMI in the 2191 days between 2010 and 2015. Table 2 shows the characteristics of the study population. The median age of the AMI patients was 66.6 years (IQR 55.6–78.2) and the age of the AMI patients were generally similar across the clusters (Cluster 1: median 66.4 years, IQR 56.1–78.1; Cluster 2: median 66.7 years, IQR 56.6–78.1; Cluster 3: median 66.8 years, IQR 56.9–78.7). The three clusters also had similar cardiovascular risk profile in terms of history of AMI/ coronary artery bypass graft (CABG)/percutaneous coronary intervention (PCI) (Cluster 1, 31.5%; Cluster 2, 32.6%; Cluster 3, 31.7%; *p* = 0.136), history of diabetes (Cluster 1, 44.2%; Cluster 2, 45.2%; Cluster 3, 44.2%; *p* = 0.127), history of hyperlipidaemia (Cluster 1, 62.3%; Cluster 2, 63.3%; Cluster 3, 63.3%; *p* = 0.211) and smoking (Cluster 1, 50.3%; Cluster 2, 50.0%; Cluster 3, 49.0%; *p* = 0.152). However, Cluster 2 had the highest proportion of patients with history of hypertension (Cluster 1, 70.9%; Cluster 2, 71.9%; Cluster 3, 70.1%; *p* = 0.008), while Cluster 3 had the highest proportion of NSTEMI patients (Cluster 1, 64.4; Cluster 2, 63.4%; Cluster 3, 66.6%; *p* < 0.001).

### 3.3. Association of Environmental Clusters with Incidence of AMI

Table 3 shows the incidence risk ratios (IRR) of AMI for Cluster 2 and Cluster 3 using Cluster 1 as reference. Compared to Cluster 1, the incidence of AMI was higher in Cluster 3 (IRR 1.04, 95% confidence interval (CI) 1.01–1.07). However, there was no significant difference in incidence of AMI between Cluster 1 and Cluster 2 (IRR 1.00, 95% CI 0.98–1.03).

The higher incidence of AMI in Cluster 3 was also observed among these subgroups: age ≥65 years (IRR 1.06, 95% CI 1.02–1.11), males (IRR 1.05, 95% CI 1.01–1.09), those without history of AMI/CABG/PCI (IRR 1.04, 95% CI 1.00–1.08), those without history of diabetes (IRR 1.04, 95% CI 1.00–1.08), those without history of hypertension (IRR 1.07, CI 1.02–1.13), those with history of hyperlipidaemia (IRR 1.05, 95% CI 1.01–1.09) and non-smokers (IRR 1.07, 95% CI 1.03–1.12). The higher incidence was apparent for NSTEMI (IRR 1.07, 95% CI 1.03–1.11) but not STEMI (IRR 1.01, 95% CI 0.96–1.06).

## 4. Discussion

In our study, we found differences in AMI risk on days classified for specific air quality and weather parameters. Higher ambient temperature and higher PSI appeared to be associated with short-intermediate increased risk of AMI, in an urban population living in a tropical Southeast Asian climate exposed to seasonal increases in air pollution.

Our findings demonstrated a potential heat effect on AMI incidence, but the existing literature appears to be diverging on the actual effect of temperature on cardiovascular outcomes. In a meta-analysis by Turner et al. (2012), the authors found no association between temperature and cardiovascular morbidity [10]. However, they excluded studies reporting only a non-linear relationship of temperature and cardiovascular morbidity as well as studies examining the effects of cold. This makes generalisability of that study difficult. Conversely, two more recent meta-analyses found both heat and cold exposure increased risk of cardiovascular hospitalisation [9,24]. This adds to the growing literature of a possible diurnal relationship of temperature and cardiovascular outcomes and different mechanisms contributing to this risk. Interestingly, geographical location and local climate may play a role in this differing effect. In the WHO MONICA project study on 24 populations around the world, they found that coronary events rates were higher during the cold periods among populations living in warmer climates, with little change in event rates in the coldest regions (Northern Sweden, North Kerelia, Kuopio) [25]. In contrast, findings by the PHEWE group (2009), conducted in 12 European cities from Mediterranean to Northern-continental countries, support a potential heat effect on cardiovascular events incidence. Although they found a negative albeit non-significant relationship between high temperatures and admissions for cardiovascular causes, they speculated that high temperatures, especially during extreme heat events, led to greater out-of-hospital cardiovascular deaths, accounting for a decrease in cardiovascular admissions [26]. This variation in event rates have been postulated to be due to behavioural changes, such as precautionary measures and acclimatisation, by the people living in different local climates [9,10,24,25]. Nonetheless, these findings were based on populations living mostly in temperate European countries and may not be reflective of the situation in other parts of the world.

From the Asian context, the effects of cold on cardiovascular outcomes seems more predominant in East Asia. In a study of three sub-tropical East Asian cities (Taipei, Kaohsiung, Hong Kong), they found that lower temperatures (<24 degrees) were associated with AMI hospitalisation, but no significant heat effect was found [27]. This was supported by another study conducted in the subtropical and tropical regions of Taiwan which found that cooler temperatures (<15 degrees) were associated with higher AMI rates [28], by using a generalised additive model to predict the onset rate of AMI. When comparing our studies to the few studies that were conducted in warmer Southeast Asia, we found that our findings of the effects of heat were mostly corroborated. A few Vietnamese studies found an acute heat effect with AMI admissions in south-central (Savanna–tropical) Vietnam (≥29.9 degrees) [29], tropical Ho Chi Min (≥31 degrees) [30] and a non-significant heat effect in the Northern Vietnamese city of Thai Nguyen (>26 degrees) [31]. Intriguingly, in cooler parts, a delayed cold effect was found in Northern Vietnam (≤16.8 degrees) [29] and in the same city of Thai Nguyen (<26 degrees; 4–15 days), with a relative risk of 1.12 for every decrease of 1 degree below threshold [31]. These temperature findings from Asia does support an apparent geographical context of temperature on cardiovascular outcomes. However, more studies are needed to better elucidate the effect of temperature, especially the heat effect, on cardiovascular outcomes in the warmer regions of Southeast Asia.

Our study found that days with higher PSI were associated with higher AMI incidence. This is largely in line with the literature of higher air pollution being associated with poorer cardiovascular outcomes [32]. However, component pollutant effect, which is what most studies examined, are less consistent. In a systemic review by Bhaskaran et al. (2009), they detected consistent effects only for PM2.5, with increasing levels associated with increased MI risk (5–7% per 10 microgram/m^3^); the effect of PM10 and gaseous pollutants (CO, NO_2_ and SO_2_) were less consistent [33]. In two other meta-analyses, one found that short term increased risk of MI was significant for all pollutants studied (CO, NO_2_, SO_2_, PM10, PM2.5) except O3 [34], while another found a pooled effect of excessive cardiovascular mortality risk only for PM2.5 (11% per 10 microgram/m^3^) [5]. When looking at low- and middle-income countries, a meta-analysis found that a 10 microgram/m^3^ increase in same-day PM2.5 and PM10 exposure was associated with a 0.47% and 0.56% increase in cardiovascular mortality respectively [35]. These studies were carried out mostly in East Asian and Pacific, Latin American and Caribbean countries and cardiovascular impact were interestingly shown to be lower than in US and European studies [4]. While these findings show that baseline air pollution exposure and levels in different regions affect cardiovascular outcomes to varying extent, higher air pollution in terms of particulate matter, which is also reflected in a higher PSI, is consistently associated with poorer cardiovascular outcomes.

Thus far, most of the studies examined the effects of weather parameters and air pollution on cardiovascular outcomes as independent variables, adjusting for each other as confounders. However, there is increasing evidence showing that air pollution and temperature may have a synergistic effect on mortality. Interestingly, these studies showed relatively more consistent results of the heat effect with pollution, particularly O_3_ and particulate matter, across both Western and Asian cities and across different climates. In a meta-analysis, a statistically significant modulation of PM10 and O_3_ by high temperatures on cardiovascular mortality was found [36]. The PHASE Project on nine cities in Europe found potential synergistic effect of temperature and O_3_ concentration on all-cause death during the warm season [37]. Another study in eight urban European countries demonstrated that high temperature enhances the effect of particle number concentration, PM2.5 and PM10 on cardiovascular mortality although they also found an enhanced cold effect on cardiovascular mortality with higher particle number concentration [38]. In two Italian studies, a synergistic effect of O_3_ and temperature with cardiovascular mortality (temperature ≥26) [39] and an increasing effect of PM10 by temperature on mortality were shown [40]. This heat effect is supported by American studies that found that particulate matter [41] and O_3_ [42] increase mortality risk on days of extreme temperature. Other studies in the subtropical climate of Brisbane found that PM10 enhances the effect of higher temperature on cardiovascular hospitalisation and mortality [43] and vice versa [44].

The modulating effect of temperature and air pollution was more varied in Asian countries. In a study of 16 metropolitan cities in Northeast Asia, a statistically significant interaction between high temperature and air pollutants (PM10, O_3_, CO, NO_2_) on mortality, with greater synergism effect of PM10 and O_3_ on cardiovascular mortality, was found [45]. In a few Chinese studies, the authors found a synergistic effect of higher temperature with PM2.5 [46,47], PM10 [48] and O_3_ [46] on circulatory mortality and both the hot and cold effect on PM10 and mortality [49]. In other parts of Asia, the cold effect on air pollution was observed in Korea (PM2.5) [50] and Hong Kong (PM10, NO_2_, O_3_) [51]. While our study did not explore the interaction between air pollution and weather, our clustering analyses identified that days with specific conditions of higher air pollution and temperature were significantly associated with higher incidence of AMI. This is largely in line with the abovementioned studies and adds to the growing literature that both air pollution and environmental parameters exert possible synergistic effect on cardiovascular outcomes.

From our subgroup analysis, we found that increased NSTEMI incidence was associated with Cluster 3 (characterized by high temperature and high PSI). This was largely in contrast to the existing literature. A Chinese study found that STEMI admission was affected by extreme high ambient temperature while NSTEMI was affected by extreme low ambient temperature [52]. Another Israeli study found significant increase of STEMI during winter with no significant seasonal variations for NSTEMI [53]. In terms of air pollution exposure, there is a lack of studies that have examined its impact on NSTEMI and STEMI occurrence separately. Most of these studies found that particulate matter exposure was associated with STEMI [54,55,56,57] and transmural infarction [58] risk but not NSTEMI. Of the few studies that found an association between NSTEMI and air pollution, their findings were not consistent. A Polish study found that PM and PM and SO_2_ were weakly associated with NSTEMI occurrence on the day of and day after exposure respectively [59]. In the English and Wales Myocardial Ischaemia National Audit Project, they found that out of four air pollutants (O_3_, NO_2_, PM 2.5, PM 10), only NO_2_ showed a positive association with NSTEMI occurrence and remained statistically significant after adjusting for PM 2.5 [60] It is well-known that STEMI and NSTEMI are fundamentally different mechanistically. STEMI occurs due to complete arterial occlusion following plaque rupture, while NSTEMI occurs due to partial vessel occlusion. Prior studies have shown that air pollutant exposure, including particulate matter and various gaseous pollutants, was associated with certain biological changes, such as levels of circulating platelet and endothelial activation markers (p-selectin, von Willebrand factor) [61], CRP and ICAM-1 [62], sCDL40 [63], platelet aggregation and thrombin generation [64] and d-dimer levels [65]. These are important markers of systemic inflammation and thrombus formation, and one may postulate that air pollutant exposure might affect thrombosis and thrombolysis to differing extents and hence STEMI or NSTEMI occurrence. Given that our study uses a composite air quality index, any effect of individual air pollutant on acute coronary syndrome (ACS) type occurrence might have been mitigated. Nonetheless, these are purely speculative observations, and more studies are warranted to elucidate the relationship and pathophysiology between weather parameters and air pollutant exposure and triggers of different ACS types.

There are a few limitations in this study. Firstly, this is an ecological study and does not allow us to prove a causal relationship between the environmental parameters and AMI incidence. Secondly, as population-level exposure and outcome were analysed, we were unable to assess the individual risk of AMI. Thirdly, we were unable to fully control for all confounding factors, including behavioural changes, socioeconomic status and ability to undertake mitigating actions in relation to weather and air pollution. For example, access to air-conditioning or being outdoors during haze period might have potential risk-modifying effects on cardiovascular outcomes. Fourthly, we are unable to examine the independent effect of constituent air pollutants on cardiovascular outcome due to the limitation of the air pollution data available to us. Previous studies have shown that PM 2.5 exposure is associated with greater myocardium susceptibility to injury as demonstrated by increased plasma cTnT concentrations [66]. Given that different pollutants may have different effects on cardiovascular outcomes, using PSI in our study could have diluted the effect size from the different pollutants. However, PSI is a readily assessable and recognisable real-world indicator of air pollution. It remains an easily understandable and interpretable tool for health education and public health policy implementation and calibration.

Our study adds to the growing literature of the impact of environmental parameters and air pollution on cardiovascular morbidity and mortality, especially in the Southeast Asian context, given the limited number of studies in this region. Environmental exposure remains an important modifiable risk factor, not only during the seasonal transboundary haze in Southeast Asia but also worldwide, with far-reaching public health implications. Quantifying the health impact of air pollution exposure, with potential day-to-day weather variations to the risk of AMI, allows for better evidence-based resource allocation. This includes potential cross-agency collaborations between meteorological and emergency services with measures including mobile app alerts, healthcare advisory and warnings, and emergency services deployment during days with heightened risks.

## 5. Conclusions

We found a transient effect of environmental parameters clustering on AMI incidence, especially NSTEMI, even after stratifying by individual characteristics. These findings have public health implications for AMI prevention and emergency health services delivery during the seasonal transboundary haze.

## Figures and Tables

**Figure 1 ijerph-19-08476-f001:**
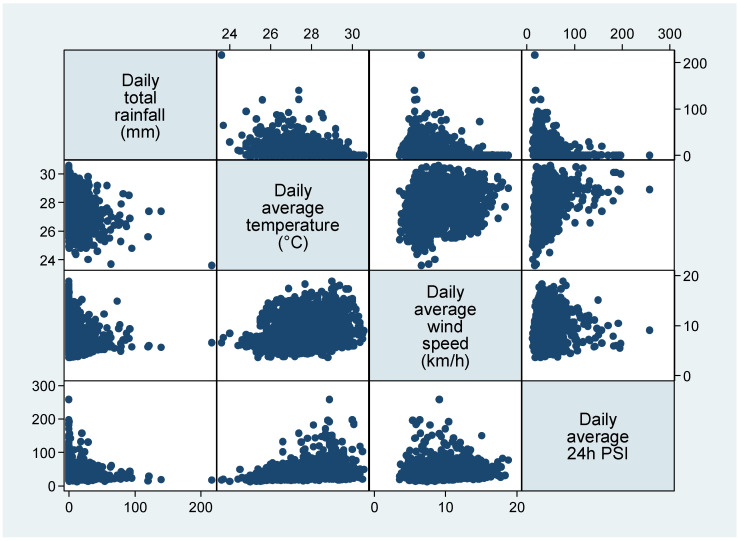
Scatterplots showing the association between the weather and air quality (N = 2191 days).

**Table 1 ijerph-19-08476-t001:** Characteristics of the clusters (N = 2191 days).

	Overall (n = 2191)	Cluster 1 (n = 686)	Cluster 2 (n = 1029)	Cluster 3 (n = 467)	*p*
**Daily total rainfall in mm, median (IQR)**	0.0 (0.0–4.0)	0.0 (0.0–0.0)	2.8 (0.4–13.6)	0.0 (0.0–2.2)	<0.001
**Daily average temperature in deg C, median (IQR)**	27.9 (27.0–28.7)	28.3 (27.6–28.9)	27.3 (26.6–28.1)	28.6 (27.8–29.0)	<0.001
**Daily average wind speed in km/h, median (IQR)**	7.8 (6.4–10.5)	10.4 (8.4–12.2)	6.6 (5.8–7.8)	8.5 (6.5–11.1)	<0.001
**Daily 24 h average PSI, median (IQR)**	32.8 (25.7–47.0)	30.5 (25.5–37.0)	29.0 (23.5–35.7)	59.2 (54.8–70.0)	<0.001

**Table 2 ijerph-19-08476-t002:** Characteristics of the AMI patients by cluster (N = 39,287 AMI episodes).

	AMI (n = 39,287)	Cluster 1 (n = 12,173)	Cluster 2 (n = 18,287)	Cluster 3 (n = 8681)	*p*
**Age, median** (**IQR**)	66.6 (56.6–78.2)	66.4 (56.3–78.1)	66.7 (56.6–78.1)	66.8 (56.9–78.7)	0.029
**Male, n** (**%**)	26,763 (68.1)	8294 (68.1)	12,417 (67.9)	5955 (68.6)	0.517
**Ethnicity, n** (**%**)					
Chinese	25,320 (64.5)	7921 (65.1)	11,717 (64.1)	5598 (64.5)	0.042
Malay	8131 (20.7)	2520 (20.7)	3784 (20.7)	1792 (20.6)	
Indian	5271 (13.4)	1586 (13.0)	2518 (13.8)	1145 (13.2)	
**Etiology, n** (**%**)					
STEMI	11,557 (29.4)	3610 (29.7)	5431 (29.7)	2482 (28.6)	<0.001
NSTEMI	25,295 (64.4)	7834 (64.4)	11,585 (63.4)	5777 (66.6)	
**History of AMI/CABG/PCI, n** (**%**)	12,584 (32.0)	3839 (31.5)	5952 (32.6)	2755 (31.7)	0.136
**History of DM, n** (**%**)	17,544 (44.7)	5379 (44.2)	8266 (45.2)	3836 (44.2)	0.127
**History of HTN, n** (**%**)	27,961 (71.2)	8630 (70.9)	13,145 (71.9)	6087 (70.1)	0.008
**History of HLD, n** (**%**)	24,726 (63.0)	7586 (62.3)	11,564 (63.3)	5492 (63.3)	0.211
**Smoking, n** (**%**)	19,258 (49.0)	6012 (50.3)	8994 (50.0)	4180 (49.0)	0.152

**Table 3 ijerph-19-08476-t003:** Incidence risk ratios of AMI across the three clusters.

	Cluster 1	Cluster 2	Cluster 3
**Number of events**	12,173	18,287	8681
**Entire cohort**	1.00	1.00 (0.98–1.03)	**1.04** (**1.01**–**1.07**) **^a^**
Subgroups			
**Age**			
<65 years	1.00	0.99 (0.96–1.02)	1.02 (0.98–1.07)
≥65 years	1.00	1.01 (0.98–1.05)	**1.06** (**1.02**–**1.11**) **^a^**
**Gender**			
Male	1.00	1.00 (0.97–1.03)	**1.05** (**1.01**–**1.09**) **^a^**
Female	1.00	1.01 (0.97–1.05)	1.03 (0.98–1.09)
**Ethnicity**			
Chinese	1.00	0.99 (0.96–1.02)	1.03 (0.99–1.07)
Malay	1.00	1.00 (0.96–1.05)	1.04 (0.98–1.10)
Indian	1.00	1.04 (0.99–1.10)	1.05 (0.98–1.12)
**Etiology**			
STEMI	1.00	1.00 (0.96–1.04)	1.01 (0.96–1.06)
NSTEMI	1.00	0.99 (0.96–1.02)	**1.07** (**1.03**–**1.11**) **^a^**
**History of AMI/ CABG/ PCI**			
Yes	1.00	1.04 (0.99–1.08)	1.05 (0.99–1.11)
No	1.00	0.99 (0.96–1.02)	**1.04** (**1.00**–**1.08**) **^a^**
**History of diabetes**			
Yes	1.00	1.03 (0.99–1.07)	1.04 (1.00–1.09)
No	1.00	0.98 (0.95–1.01)	**1.04** (**1.00**–**1.08**) **^a^**
**History of hypertension**			
Yes	1.00	1.02 (0.99–1.05)	1.03 (0.99–1.07)
No	1.00	0.97 (0.93–1.01)	**1.07** (**1.02**–**1.13**) **^a^**
**History of hyperlipidemia**			
Yes	1.00	1.02 (0.98–1.05)	**1.05** (**1.01**–**1.09**) **^a^**
No	1.00	0.98 (0.94–1.02)	1.02 (0.98–1.07)
**Current/former smoker**			
Yes	1.00	1.00 (0.97–1.03)	1.02 (0.98–1.06)
No	1.00	1.01 (0.97–1.04)	**1.07** (**1.03**–**1.12**) **^a^**

^a^ significant result *p* < 0.05.

## Data Availability

Not applicable.

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
