# Peer review of "Clustering of Environmental Parameters and the Risk of Acute Myocardial Infarction"

_ijerph, 2022, doi:10.3390/ijerph19148476_

Round 1

Reviewer 1 Report

Dear Authors,

The article is very interesting. However, to improve the quality of the article, I propose additional analysis regarding the effect of air pollutants on AMI risk.

The cluster division adopted by the Authors distinguishes three clusters, one of which is associated with an increased risk for AMI. Clusters 1 and 2 have higher wind speed and precipitation frequency, respectively. These are factors that directly affect the lower concentration of pollutants in the air.  
The authors show that an important factor that influences the increase in AMI risk is temperature, whose average values are highest for cluster 3. 

Therefore, in my opinion, the influence of temperature is not clear and requires additional analysis because the Pollutant Standards Index (PSI) is the most significant. 
What is the relationship between only temeprature and PSI (for the same wind speed and rainfall conditions)?

It would be useful to clarify the value of the air quality index. Which of the 6 pollutants analyzed had the highest concentration and greatest impact on the PSI value?

In summary, it would be beneficial to complete information regarding:

1) What pollutants were present in the air at the highest concentrations relative to PSI air quality standards.

2) Highlight the meteorological conditions where the greatest impact on PSI and associated AMI risk was observed.

Best regards

Author Response

Dear Reviewer,

Please kindly refer to the attachment.

Thank you.  

Reviewer 2 Report

The authors investigated the impact of aerial pollution on acute myocardial infarction risk in Singapore. They used a form of cluster analysis to investigate the correlation of time segments characterized by specific environmental conditions with incidence of myocardial infarction, under consideration of different population parameters and co-morbidity.

The English of the text is good and only a minor editorial check for spelling and grammar may be sufficient. However, this text was not written by a native speaker and there are some minor language related points that have been outlined below.

The methodology of the manuscript appears to be adequate, cluster analysis represents a common statistical tool for investigating environmental short term health effects. Results are presented well in tabular form and in text. The effects observed are small (mean IRRs below 1.1), but plausible and thus informative.

The detected high(er) IRR for NSTEMI in Cluster 3 is indeed interesting. As this is a quite significant finding of this paper, I would recommend to shortly provide a little more information on the diagnostic method. Is is it correct that NSTEMI diagnosis was mainly based on cardiac troponin concentrations? What additional diagnostic steps were included to diagnose AMI-NSTEMI?

There are some publications discussing a possible correlation between NSTEMI and particulate matter (PM2.5, PM10) in the environment. Is it possible that Cluster 3 is also a cluster with high particulate matter (i.e. dust linked to temperature)? There are also publications that indicate a specific increase of cardiac troponin due to environmental particulate matter (Environ Pollut. 2021 Apr 15;275:116663. doi: 10.1016/j.envpol.2021.116663). This should be added to discussion and literature.

Minor points:

l. 98: Language: "The outcome of this study is the occurrence of AMI;"

l. 102: Language: "We excluded inpatient cases as AMI that occurred while patients were hospitalised for non-AMI conditions might have pathophysiologic origins that differ from outpatient cases. " This sentence should be corrected and made more clear.

l. 116: "The readings are measured from 5 meteorological stations"

l. 136: ".. will be excluded..." Consistent time form has to be used in text.

l. 189: "In our study, we found that air quality and weather parameters could classify days

into 3 distinct clusters, with differences in AMI risks between them." This sentence needs to be rearranged like: "In our study we found differences in AMI risk at days classified for specific air quality and weather parameters."

l. 195: "..varied..", probably better: "diverging".

l. 232: "These finding does..."

l. 240: better: "..., they detected consistent effects only for PM2.5"

l. 255: Language: "Thus far, the evidence presented studied the independent effects of meteorological parameters and air pollution with cardiovascular outcomes..."

Author Response

Dear Reviewer,

Please kindly refer to the attached document.

Thank you.

Reviewer 3 Report

The paper is poorly written and scientifically uncorrect. Lots of variables should be included in order to definetly assess the correlation between "air" and "AMI". There are no data about pharmacological treatments, comorbidities, type of AMI, biochemical parameters, etc. Different areas and different air conditions might impact on results. Pollution data had not been considered. This seems rather a study based on a poor statistical correlation but with no solid data

Author Response

(The authors gave the same response as above.)
